# Exploring the effect of menstrual loss and dietary habits on iron deficiency in teenagers: A cross-sectional study

Lisa Söderman[1,2]*, Anna Stubbendorff[3], Linnea V. Ladfors[4],
Beata Borgström Bolmsjö[5,6], Peter Nymberg[5,6], Moa Wolff[5,6]

1 Department of Clinical Science and Education, Södersjukhuset, Karolinska Institutet, Stockholm, Sweden, 2 Department of Obstetrics and Gynecology, University of British Columbia, Vancouver, Canada, 3 Department of Clinical Sciences Malmö, Lund University, Malmö, Sweden, 4 Division of Clinical Epidemiology, Department of Medicine, Solna, Karolinska Institutet, Stockholm, Sweden, 5 Center for Primary Health Care Research, Department of Clinical Sciences, Lund University, Malmö, Sweden, 6 Office for Primary Care, Skåne University Hospital, Lund, Sweden

* lisa.soderman@ki.se

## Abstract

Adolescent girls are particularly susceptible to iron deficiency due to increased iron requirements during the pubertal growth spurt, combined with iron loss following menarche. This study aimed to investigate the prevalence of heavy menstrual bleeding in an adolescent population using the SAMANTA questionnaire and to explore its relationship with dietary habits and iron deficiency. This cross-sectional study was conducted in two Swedish high schools in 2023. Post-menarchal female students, aged 15 and older, were included (n = 394). Data were collected on-site through a patient-reported survey, including the SAMANTA questionnaire for heavy menstrual bleeding, and by blood sampling. Meat-restricted diet was analyzed in relation to iron status. Descriptive analysis and regression analysis were used to assess the prevalence of heavy menstrual bleeding and its relationship with dietary habits and iron deficiency, defined as ferritin <15 µg/L. The prevalence of heavy menstrual bleeding and iron deficiency in the cohort was 53% (208/394) and 40% (157/394), respectively. In univariate analysis, heavy menstrual bleeding (OR 3.0, 95% CI [2.0, 4.6]) and a meat-restricted diet (OR 3.5, 95% CI [2.2, 5.6]) were both associated with increased odds of iron deficiency. When assessing the joint effect of having heavy menstrual bleeding and a meat-restricted diet, the odds of iron deficiency were 13.5 times higher compared to omnivore individuals with normal menstruation (OR 13.5, 95% CI [6.4, 28.7]). Overall, the prevalence of iron deficiency in this population of adolescent girls was very high. Heavy menstrual bleeding and a meat-restricted diet were both independently associated with increased odds of iron deficiency. However, odds for iron deficiency were monumentally higher when combining these two variables, thus

**Data availability statement:** Due to the regulations provided following the ethics approval by the Swedish Ethical Review Authority, it is not possible to make individual-level data publicly available. The participants were informed that results would only be published at the group level and cannot be traced back to individuals. Providing data freely available to other researchers would make it possible to trace individual data and would breach compliance with the approved protocol. The data supporting the findings of this study are available from the Swedish National Data Service (SND) repository. Access to the dataset will be granted upon reasonable request by contacting request@snd.se.

**Funding:** Open access funding provided by Karolinska Institutet. This study was funded by the Southern Health Care Region of Sweden, the Lions Research Fund Skåne, and Regional Funding for Clinical Research (USVE), awarded to MW. A grant from the Vetenskapsrådet (2023-06565) funded the participation of LS. The funders had no role in study design, data collection and analysis, decision to publish, or preparation of the manuscript.

**Competing interests:** Lisa Söderman and Moa Wolff received an honorarium for an educational webinar by Pharmacosmos. This does not alter our adherence to PLOS ONE policies on sharing data and materials.

highlighting the importance of assessing and addressing both excessive output and low intake of iron.

## Introduction

Adolescent girls are particularly susceptible to iron deficiency (ID) due to increased iron requirements during the pubertal growth spurt, combined with iron loss following the onset of menstruation. A sufficient intake is crucial for maintaining adequate levels of iron. When iron loss consistently exceeds intake over time, it can lead to ID and ID anemia. Even before anemia develops, ID is associated with symptoms such as muscular weakness [1], reduced physical performance [2], depressive symptoms [3], fatigue [4], and impaired cognitive function [5,6]. These symptoms are reversible when the iron stores are replenished [7].

The definition of heavy menstrual bleeding (HMB) has evolved over time. Historically, it was defined as menstrual bleeding lasting more than seven days [8], or blood loss exceeding 80 ml [9]. The prospective pictorial blood assessment chart (PBAC) estimates the volume of menstrual blood loss by measuring the saturation of each tampon and sanitary pad [10]. While accurate, it is inconvenient for routine clinical use. Today, HMB is understood as menstrual loss that is of sufficient volume to adversely impact a woman's physical, emotional, social, and/or material quality of life (QoL) [11]. The newly developed six-item SAMANTA questionnaire, which takes these measures into account, has proven to be a valid and accessible tool for identifying HMB in adult women, compared with PBAC [12].

HMB in adolescents is often due to anovulatory menstrual cycles [13]. However, approximately 22% of adolescents with HMB presenting at a tertiary healthcare center had a bleeding disorder, such as von Willebrand disease or inherited platelet function disorder, which could also contribute to the excessive bleeding [14].

Comparisons of ID prevalences are challenged by varying cutoffs for ferritin levels, but it is estimated to be the most prevalent nutritional deficiency globally [15]. A ferritin level below 15 µg/L is defined as ID in adults and adolescents according to the World Health Organization (WHO) [16]. The clinical value of this cutoff is debated [7]. A 2022 Delphi study recommended using <20 µg/L as the ferritin cutoff in pediatric populations and <30 µg/L for adults, as well as for children with risk factors for ID [17].

There is a knowledge gap regarding the impact of HMB and dietary iron intake on ID in adolescents. Additionally, there is a lack of accurate and accessible tools for screening for HMB in this age group, as the SAMANTA questionnaire has not previously been validated for use in adolescents.

The aim of this study was to investigate the prevalence of HMB in an adolescent population and examine its relationship with dietary habits and ID, as well as to evaluate the applicability of the SAMANTA questionnaire in this age group.

## Materials and methods

The Iron Insight study was a cross-sectional study conducted at two high schools in southern Sweden in October 2023. Invitations for participation in the study were made through advertisements posted in the schools, on the school web homepage,

and through information disseminated by teachers. On two separate occasions at each school, October 2 and 6 in Lund and October 9 and 16 in Malmö, all individuals aged 15 years or older who had experienced menarche were invited to attend a general information session in the main hall, where study personnel provided information about the study as part of the recruitment process. Those who wanted to participate were provided with a QR code linked to an informed consent document that was accessible via mobile phone or computer. No parental consent is required for participants 15 years of age or older. If informed consent was obtained, the participants were forwarded to a digital REDCap questionnaire (S1 File Questionnaire) [18,19].

Following completion of the questionnaire, the participant´s weight and height were measured, and the study personnel drew blood samples for hemoglobin and ferritin. Participants were awarded a light snack and a cinema gift card for taking part.

The inclusion criteria were post-menarchal females, aged 15 or older. Participants reporting current pregnancy, bacterial infection, and/or chronic inflammatory diseases such as inflammatory bowel disease and rheumatoid arthritis were excluded. Participants using hormonal contraceptives were excluded from the main analysis due to the modifying effect these have on menstrual bleeding patterns [20] and due to inconsistencies in questionnaire responses. This is explained in more detail in the results section.

Characteristics including age, age at menarche, weight, height, use of contraceptives, bleeding pattern while using hormonal contraceptives, iron-supplement intake, smoking, and use of snus were collected.

The intake of iron supplements was assessed in relation to menstruation and dietary preferences. Information on dietary habits, reflecting the intake of meat and thereby heme iron, during the previous year was collected using selected items from the questionnaire by Stubbendorff et al [21]. Participants self-classified as either *omnivores* (no dietary restrictions) or as following a *meat-restricted diet*, which included those who avoided red meat, followed a pescatarian diet, or were vegetarian or vegan. A meat-restricted diet was used as a proxy for low dietary intake heme iron. Parts of this data have been presented previously in "Iron insight: exploring dietary patterns and iron deficiency among teenage girls in Sweden" by Stubbendorff et al [21].

HMB was assessed through the validated SAMANTA questionnaire [12]. The questions were: (1) Do you experience menstrual bleeding for more than seven days per month? (2) Do you experience three or more days of heavier menstrual bleeding during your menstrual period? (3) In general, does menstruation bother you due to its abundance? (4) During any of these heavier menstrual bleeding days, do you spot your clothes at night, or would you spot them if you did not use double protection/did not change your clothes during the night? (5) During these heavier menstrual days, are you worried about staining the chair, sofa, etc.? (6) In general, during these heavier menstrual bleeding days, do you avoid, as far as possible, some activities, trips, or leisure-time plans because you frequently need to change your tampon or sanitary towel? Affirmative answers to items (1) and (3) were assigned a score of three points each, and the other questions were assigned a score of one point each. A total score of three points or more was considered HMB [12]. As no Swedish version was available, the six SAMANTA questions were first translated from English into Swedish by a Swedish-speaking researcher and general practitioner. A professional English editor then performed a back-translation into English. The Swedish translation was subsequently reviewed and adjusted to reflect the original phrasing and intent best. This process followed the principles of forward–backward translation. Self-perceived general health was assessed with a stand-alone question from the KIDSCREEN-10 questionnaire (how would you describe your general health: poor, fair, good, very good, excellent) [22]. The poor and fair values were grouped as poor general health, and good, very good, and excellent health were categorized as good general health (S1 File Questionnaire).

Registered nurses collected non-fasting blood samples (6–8 ml) which were stored in +2 to +8°C overnight, transported to and analyzed at the regional clinical chemistry laboratory in Skåne (Lund and Malmö) the following day. Blood samples for ferritin were centrifuged within four hours of phlebotomy.

Levels of ferritin and hemoglobin were analyzed using the Atellica IM Analyzer [23] and Sysmex XN-10 [24], respectively.

Serum ferritin lower than 15 μg/L was used as a cut-off for ID, and hemoglobin lower than 120 g/L for anemia [25]. For secondary analysis, ferritin<30 μg/L was tested as a cutoff, as this is the recommended clinical cutoff for ID according to the 2022 Delphi study [17].

Within 2–3 days, all participants received an email with their lab results and general information about available resources for addressing HMB, low-iron diet, fatigue, and suspected psychiatric symptoms. Participants with anemia (Hb < 110 g/L for ages 10–18 or Hb < 117 g/L for age > 18 as per local reference values) [26] or with markedly elevated ferritin levels (>300 μg/L, indicating possible iron overload requiring medical assessment) were referred to their primary care center for further evaluation and follow-up.

The non-anemic participants with iron deficiency were advised to take an iron supplement for 3–6 months, along with information about diet and menstruation to help reduce the risk of future iron deficiency.

The study design and procedures were approved by the Swedish Ethical Review Authority, nr. 2023-01088-01. Participants provided written informed consent before inclusion in the study.

The study was conducted according to the STROBE guidelines [27].

## Statistical analysis

Descriptive data are presented as numbers, proportions, and continuous variables as medians with interquartile ranges (IQR) or means with standard deviations (SD). Analysis comparing continuous variables was made using the Mann-Whitney U test or Kruskal-Wallis test for variables with asymmetrical distribution and an independent t-test for normally distributed variables. The $\chi^2$ test or Fisher´s exact test was used for categorical variables.

Binary logistic regression was used to evaluate the association between each SAMANTA item and ID as well as for HMB, a meat-restricted diet, and ID. BMI and years of menstruation (age – age at menarche) were included in the multivariable analysis as continuous variables.

Confounders of the association between HMB and ID, i.e. that covariates that could influence both the risk of HMB and of ID, were selected based on the directed acyclic graph framework and included: BMI and years with menstruation (S1 Fig) [28,29]. A higher BMI is associated with heavy menstrual bleeding in young women [30]. Obesity as well as a low BMI is associated with iron deficiency [31]. Participants with fewer years of menstruation are at higher risk of heavy anovulatory bleeding which is more common in the years after menarche. Study participants with younger age at menarche will have had more years with menstruation and are thus at risk of negative iron balance due to loss. The use of iron supplements and dietary restrictions were not considered confounders of the relationship between HMB and ID as per S1 Fig.

Results were presented as odds ratios (OR) with 95% confidence intervals (95% CI) for both crude models and models adjusted for BMI and years of menstruation.

Due to the increased risk of ID associated with meat-restricted diets, diet was treated as an effect modifier of the relationship between HMB and iron deficiency. Consequently, the regression analysis was stratified by dietary group (omnivorous vs. meat-restricted).

All hypothesis testing used a 2-tailed $p < 0.05$ as the significance level and 95% CI. Of the analyzed cohort, data were missing for weight (n = 66, 16.8%), height (n = 41, 10.4%), iron supplement (n = 1, 0.3%), age at menarche (n = 3, 0.6%), and hemoglobin (n = 1, 0.3%). Multiple imputation was used to account for the missing data of BMI, 66 (16.8%), for cases who opted out of being weighed and/or measured. Other missing data were deleted on a per-analysis basis. In a sensitivity analysis, we reran the analysis of the association between HMB/diet and ID in the sample with complete BMI data (complete case analysis) and compared the estimates with those from the whole sample.

The Statistical Package for Social Sciences software (IBM® SPSS® version 29 for Windows; IBM Corp.) was used for statistical analyses.

## Results

A total of 584 participants were screened, 544 were eligible, and 477 were included. Amenorrhea due to hormonal contraceptive use was reported by seven participants, three of whom also reported HMB, suggesting inconsistent SAMANTA scores. The most prevalent contraceptives were combined oral contraceptive pills (58/477; 12.2%), followed by etonogestrel subdermal implant (13/477; 2.7%), levonorgestrel intrauterine system (9/477; 1.9%), and progestin-only pills (5/477; 1.0%).

Participants using hormonal contraceptives (82/477; 17.2%) or who did not provide information on contraceptive use (1/477; 0.2%) were excluded from the analysis. The flowchart of the inclusion process is shown in Fig 1.

The final cohort included 394 participants. The median age was 16.0 (IQR: 16.0–17.0), and the median BMI was 21.2 (IQR: 19.4–23.2). The reported median age at menarche was 12.0 years (IQR 12.0–13.0). A total of 27.9% (n = 110) reported following a meat-restricted diet (Table 1).

More than half of the participants reported HMB, i.e., SAMANTA score ≥3 (52.8%, 208/394). Worrying about staining chairs, sofas etc. was reported by 66.8% (263/394). Avoiding activities, trips, or leisure-time plans was reported by 37.6% (148/394) (Fig 2). The odds of having iron deficiency associated with each individual item from the 6-item SAMANTA questionnaire is shown in Fig 2. Participants with HMB reported menarche at a younger age than those without HMB (p = 0.005; Table 1).

The cumulative probability of ID was consistently higher among participants with HMB across all cutoff values of ferritin (Fig 3).

We observed no difference in self-rated health (according to KIDSCREEN-10) between the groups with and without HMB (S1 Table).

Iron supplement use was not more common among participants with HMB; 6.8% (14/207) reported using supplements, compared to 7.5% (14/186) among those without HMB (p = 0.769; Table 1). In contrast, iron supplement use was significantly more frequent among participants with a meat-restricted diet (11.6%, 15/129) than among omnivores (5.6%, 20/355, p = 0.024), with the highest proportion observed in the vegan/vegetarian subgroup (S2 Table).

ID was highly prevalent in the study population. Overall, 39.8% (157/394) of participants had ID, defined as ferritin <15 µg/L, and 73.6% (290/394) had ferritin levels below 30 µg/L.

Both ID and anemia were significantly more common among participants with HMB than among those with normal menstrual bleeding. Among participants with HMB, 51.9% (108/208) had ferritin <15 µg/L, 79.8% (166/208) had ferritin <30 µg/L, and 12.5% (26/208) were anemic. In comparison, among those without HMB, 26.3% (49/186, p < 0.001) had ferritin <15 µg/L, 66.7% (124/186, p = 0.003) had ferritin <30 µg/L, and 4.3% (8/186, p = 0.004) were anemic (S3 Table). Median ferritin levels were significantly lower in participants with HMB (14.0 µg/L, IQR 8.0–25.8) compared to those without HMB (22.0 µg/L, IQR 14.0–37.0; p < 0.001). Hemoglobin concentrations were also lower among participants with HMB (131.0 g/L, IQR 124.0–138.0) than those without HMB (134.0 g/L, IQR 127.0–139.0; p < 0.001; S3 Table).

The ferritin levels were low across all dietary groups. When dietary habits were assessed together with menstrual bleeding patterns, the highest median ferritin level was observed among omnivores with normal menstruation (26.0 µg/L; 95% CI: 24.0–30.0), and the lowest among participants with both HMB and a meat-restricted diet (11.0 µg/L; 95% CI: 8.0–13.0; Fig 4). ID was most prevalent among participants with both HMB and a meat-restricted diet, affecting 70.9% (39/55) in this group. Among omnivores with HMB, 45.1% (69/139) were iron-deficient (S3 Table).

Univariate logistic regression analysis showed that both HMB and a meat-restricted diet were independently associated with increased odds of ID. HMB was associated with a three-fold increase in odds (OR 3.0; 95% CI:2.0–4.6), while the odds were similarly elevated among those following a meat-restricted diet (OR 3.5; 95% CI: 2.2–5.6; Fig 5). There was virtually no shift in estimates after adjustment for BMI and years of menstruation.

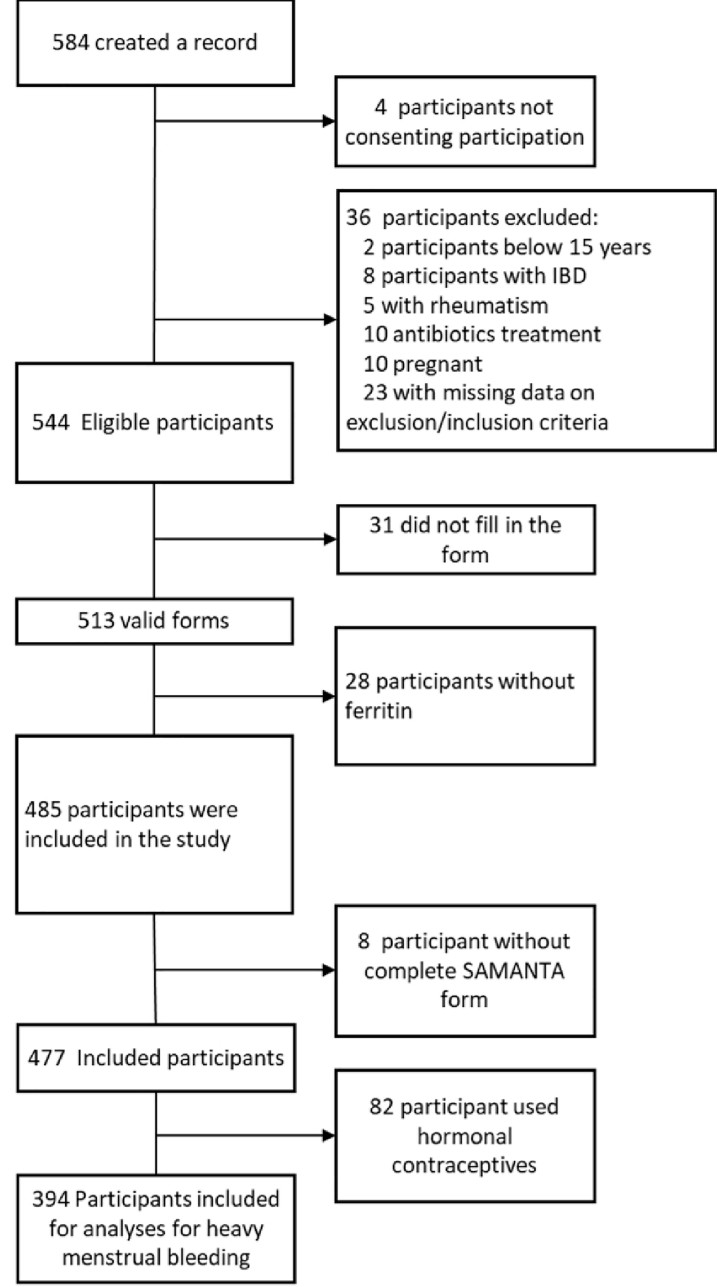

**Fig 1. Flowchart of the inclusion process.**

An additive association was observed when examining the combined effect of these two risk factors. Participants with both HMB and a meat-restricted diet had the highest odds of ID, with more than a fourfold increase compared to either factor alone (OR 13.5; 95% CI: 6.4–28.7), using omnivores without HMB as the reference group. This effect was somewhat attenuated after adjustment for BMI and years of menstruation (Fig 5). This association remained when using the higher ferritin cutoff (< 30 µg/L), although the effect sizes were less pronounced (S4 Table).

**Table 1. Characteristics of participants in the study population, overall and by heavy menstrual bleeding (HMB) status.**

| | All<br>n = 394 | HMB<br>n = 208 (52.8%) | Non- HMB<br>n = 186 (47.2%) | |
|---|---|---|---|---|
| | median (IQR)/<br>mean (±SD) | median (IQR)/<br>mean (±SD) | median (IQR)/<br>mean (±SD) | p-value |
| Age, in years | 16.0 (16.0-17.0) | 16.0 (16.0-17.0) | 16.0 (16.0-17.0) | 0.083 |
| Height, in cm | 168.1 (±6.5) | 167.9 (±6.4) | 168.4 (±6.6) | 0.464 |
| Weight, in kg | 60.2 (54.9-66.9) | 60.6 (55.7-67.8) | 59.6 (54.4-65.9) | 0.290 |
| BMI | 21.2 (19.4-23.2) | 21.6 (19.4-23.5) | 20.9 (19.4-23.0) | 0.149 |
| Age at menarche, in years | 12.0 (12.0-13.0) | 12.0 (11.0-13.0) | 13.0 (12.0-13.0) | 0.005 |
| Years with menstruation | 4.0 (3.0-5.0) | 4.0 (3.0-5.0) | 4.0 (3.0-5.0) | 0.156 |
| | n (%) | n (%) | n (%) | |
| **Menarche** | | | | |
| Age 11 years or younger | 83 (21.2%) | 53 (25.7%) | 30 (16.2%) | 0.020 |
| Age 12 or 13 | 242 (61.9%) | 126 (61.2%) | 116 (62.7%) | |
| Age 14 or older | 66 (16.9%) | 27 (13.1%) | 39 (21.2%) | |
| **Geographic area of residence** | | | | |
| City | 284 (72.1%) | 147 (70.7%) | 137 (73.7%) | 0.464 |
| Smaller town | 88 (22.3%) | 51 (24.5%) | 37 (19.9%) | |
| Rural area | 22 (5.6%) | 10 (4.8%) | 12 (6.5%) | |
| **Smoking** | | | | |
| No | 287 (72.8%) | 153 (73.6%) | 134 (72.0%) | 0.547 |
| Yes, sporadically | 100 (25.4%) | 50 (24.o%) | 50 (26.9%) | |
| Yes, daily | 7 (1.8%) | 5 (2.4%) | 2 (1.1%) | |
| **Snuff** | | | | |
| No | 334 (84.8%) | 177 (85.1%) | 157 (84.4%) | 0.303 |
| Yes, sporadically | 33 (8.4%) | 20 (9.6%) | 13 (7.0%) | |
| Yes, daily | 27 (6.9%) | 11 (5.3%) | 16 (8.6%) | |
| **Dietary preferences** | | | | |
| Omnivore | 284 (72.1%) | 153 (73.6%) | 131 (70.4%) | 0.626 |
| No red meat | 23 (5.8%) | 13 (6.3%) | 10 (5.4%) | |
| Pescetarian | 31 (7.9%) | 13 (6.3%) | 18 (9.7%) | |
| Vegan/Vegetarian | 56 (14.2%) | 29 (13.9%) | 27 (14.5%) | |
| **Dietary preferences – binary** | | | | |
| Omnivore | 284 (72.1%) | 153 (73.6%) | 131 (70.4%) | 0.490 |
| Meat-restricted diet [a] | 110 (27.9%) | 55 (26.4%) | 55 (29.6%) | |
| **Takes iron supplement** | | | | |
| No | 365 (92.9%) | 193 (93.2%) | 172 (92.5%) | 0.769 |
| Yes | 28 (7.1%) | 14 (6.8%) | 14 (7.5%) | |

[a] Meat-restricted diet includes no red meat, pescetarian and vegan/vegetarian.

Participants with anemia (hemoglobin <120 g/L; n = 37) were more likely to have HMB ($p = 0.004$) and a lower BMI ($p = 0.042$) compared to those without anemia. However, no significant differences were found between anemic and non-anemic participants regarding dietary restrictions (S5 Table).

A sensitivity analysis was conducted to assess the robustness of the multiple imputations for missing BMI data. A complete case analysis was performed (n = 328), and the logistic regression showed slightly higher odds ratios for ID when

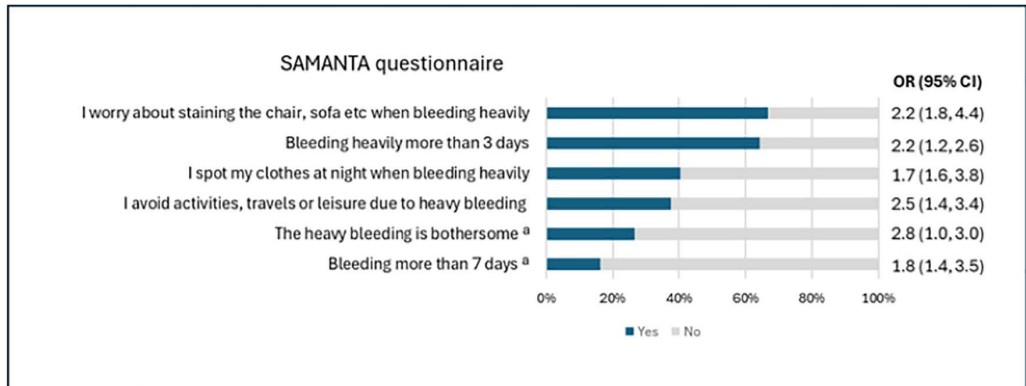

**Fig 2. Description of the proportions of affirmative answers in the SAMANTA questionnaire and odds ratio for iron deficiency for each question, univariate analysis.** ᵃQuestions that were assigned three points each, all other questions were assigned one point each if affirmative answer. n = 394. Abbreviated questions for the figure, full questions are available in the method section.

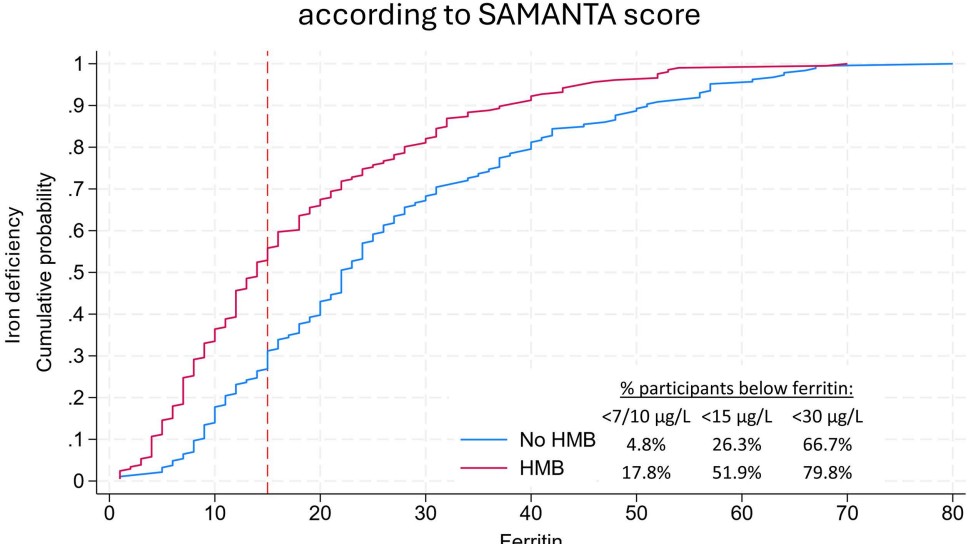

**Fig 3. Cumulative distribution functions of ferritin levels and probability of iron deficiency, stratified by heavy menstrual bleeding (HMB) defined as SAMANTA score ≥3.**

assessing the influence of dietary restrictions alone and in combination with HMB, compared to the main analysis. However, overlapping CIs indicated that these differences were not significant (S6 Table).

## Discussion

In this cross-sectional study of adolescent girls, the prevalence of ID was alarmingly high: 40% had ferritin levels below the WHO-defined cutoff for ID. Over half of the participants reported HMB according to the newly developed SAMANTA questionnaire, which was strongly associated with lower ferritin and hemoglobin levels. The lowest values were observed among girls with both HMB and a meat-restricted diet.

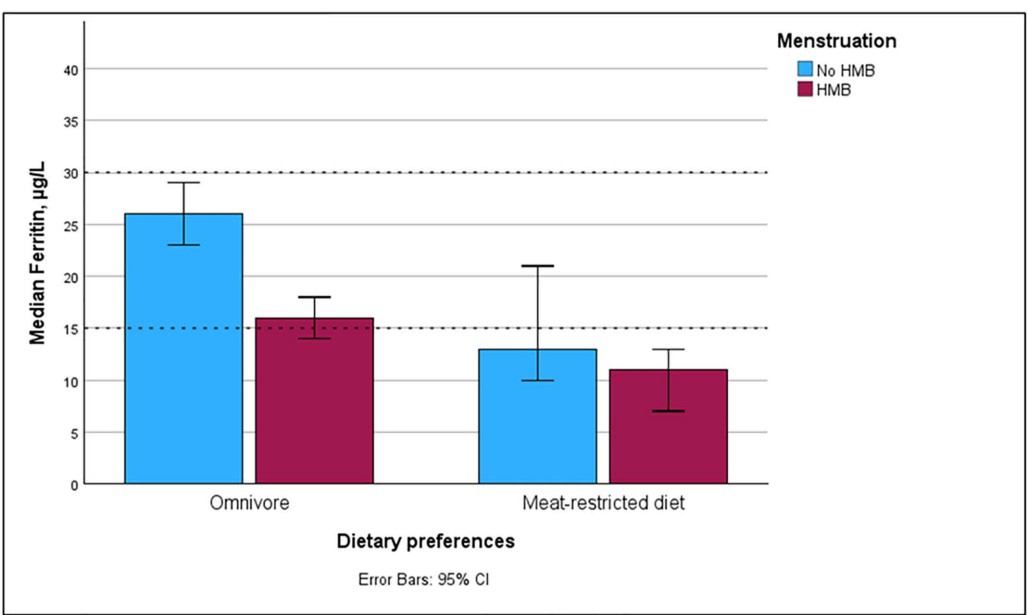

**Fig 4. Median ferritin in women with and without heavy menstrual bleeding (HMB), stratified by diet (omnivore vs meat restricted).**

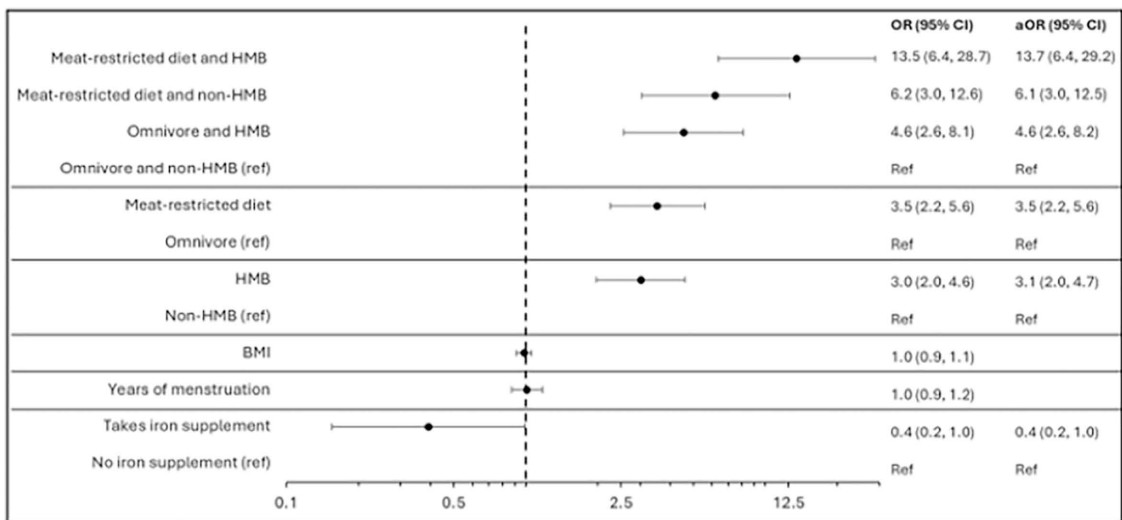

**Fig 5. Forest plot showing crude odds ratios (OR) with 95% confidence interval (CI) for iron deficiency (ID) (ferritin <15 µg/L) from logistic regression analysis. OR, crude and adjusted for BMI and years of menstruation are presented next to the plot.**

Regression analyses confirmed that HMB and a meat-restricted diet were independently associated with an increased risk of ID, each roughly tripling the odds. Notably, the combination of these two factors resulted in a more than fourfold additive increase, with the odds of ID rising over 13 times compared to participants with neither risk factor – an association that remained significant after adjusting for BMI and years of menstruation. This result may have been expected, yet the magnitude of the combined effect has not been previously reported and deserves clinical attention.

Our data are in line with previous Swedish prevalence studies for ID in girls, which showed a 44% prevalence for ID in 16-year-olds in the year 2000 [32], and a prevalence of 30% in 15-year-olds in 2016 [33]. Considering the high prevalence of ID found in our study, many adolescents may experience reversible symptoms such as physical fatigue and cognitive impairment, potentially affecting academic performance during these formative years.

The prevalence of HMB of 53% is in line with previous studies that used the PBAC [10] in populations of similar age [34,35]. This could suggest that the SAMANTA questionnaire is an alternative to the, albeit prospective, but more demanding, PBAC. While previous studies have shown that HMB is associated with lower QoL [36,37], we could not confirm this in our study, but the bleeding seemed to put a hindrance on the daily lives of 38% of the participants who reported avoiding certain activities during days of heavy bleeding.

Screening for HMB and identifying individuals at increased risk for ID in schools may lead to an earlier diagnosis of ID and anemia.

The dietary patterns of this cohort have previously been presented in detail, with adjustment for HMB [21]. However, menstrual factors and their association with iron deficiency were not analyzed in depth in that publication. The dietary intake of iron has been shown to be lower than the recommended daily intake in most adolescent girls in Sweden [33]. The intake of meat, and thus of heme iron, which has a higher absorption rate than the non-heme iron in plant-based products, is decreasing among Swedish women [38]. A reduced intake of red meat was advised in the most recent Nordic Nutritional Recommendations [39] due to both health and environmental reasons. As shown by Labba et al [40], many ready-made meat substitutes contain a high level of phytates, which impairs the absorption of declared non-heme iron content. The absorption of non-heme iron can be significantly enhanced by co-consuming vitamin C or other organic acids [41]. This highlights the need for information and education on how to ensure sufficient iron intake and optimize its absorption.

Our study found that young females adhering to meat-restricted diets report use of iron supplements more frequently than those experiencing HMB. This may reflect a greater awareness of iron deficiency risk among individuals on plant-based diets, who proactively compensate through supplementation. Hence, while these individuals take preventive action, adolescents with HMB seem to underestimate the substantial iron loss associated with heavy periods, which is possibly due to difficulties in accurately assessing menstrual blood loss. This underestimation can lead to a lack of perceived need for supplementation, despite their increased risk of ID. Addressing this gap requires targeted education to raise awareness among young women with HMB about the importance of monitoring menstruation and maintaining adequate iron intake. Oral iron supplements are sold over the counter in Sweden and are considered safe when taken according to recommendations. However, gastrointestinal side effects are common, which often leads to low compliance [42].

Many of these iron-deficient individuals may become pregnant in the near future, putting them at increased risk for pregnancy-related complications. ID during pregnancy has been associated with irreversible impairments in cognitive and motor development, premature delivery, intrauterine growth restriction, maternal anemia and an elevated risk of postpartum hemorrhage [43].

A key strength of this study was the possibility of combining information on menstrual status with dietary habits for a more extensive view of iron balance in this population. A limitation of the study is that the results may be affected by self-selection bias and recall bias. The temporal aspect of which menstrual period was reported in the SAMANTA questionnaire was not specified, resulting in some conflicting answers amongst participants with hormonal contraceptives who reported amenorrhea but scored high on the SAMANTA score, probably describing menstruation prior to the use of hormonal therapy. To reduce bias, participants with hormonal therapy were therefore excluded from the analysis. However, the prevalence of HMB is in line with previous studies, suggesting the accuracy and generalizability of the results among an adolescent population.

Girls identified as having HMB by the SAMANTA questionnaire had lower ferritin levels across all iron-deficiency cutoffs compared to those without HMB, supporting the validity of the tool in this population. The individual items of the

SAMANTA questionnaire clearly distinguished between participants with and without iron deficiency. However, the current scoring system may require revision when applied to adolescents, as the items most strongly associated with iron deficiency in this group appear to differ from those in adults and premenopausal women.

As we did not collect data on eating disorders, stratifying dietary preferences solely by meat consumption (omnivore vs. meat-restricted) may overlook important underlying factors associated with restrictive eating patterns. We had no information regarding physical activity, which may be associated with reduced iron-absorption [44]. No inflammatory markers were collected, which could have led to the inclusion of participants with inflammatory disease who may have inflated ferritin levels but still suffer from ID. Information on the duration and amount of iron supplements taken was not available. Bodily measurements, height, and weight were not obtained in 66 participants (16.8%), probably due to a lack of privacy during data collection. Lastly, by design, cross-sectional studies do not allow identification of causal effects, and residual confounding cannot be ruled out.

A systematic review by Truong et al (2024) [26] showed a discrepancy between lab reference values and clinical guidelines regarding the cutoff for iron deficiency. The reference values for most commercial ferritin assay laboratories globally are set at lower than ferritin 15 µg/L [45], including the lab used for this study, which has ferritin lower than 7 µg/L as a reference value for the study population. This means that the lab results for ferritin are marked as normal even when they are lower than 15 µg/L, leading clinicians to unknowingly sustain underdiagnosis and undertreatment of ID. A harmonization of the reference values from laboratories with the recommended cutoff values in clinical guidelines is needed to ensure accurate diagnostics of ID. Interestingly, these laboratory cutoffs are set lower for females than for males, suggesting lower physiological iron requirements in females, which runs counter to scientific evidence [45].

## Conclusion

ID was highly prevalent in this population of adolescent girls. Both HMB and a meat-restricted diet were independently linked to increased odds of ID, and when combined, they were associated with more than a 13-fold increase in these odds. These findings underscore the importance of identifying and addressing both iron loss and low iron intake when screening for and preventing ID in adolescents. The SAMANTA questionnaire, which was validated for use in this age group as part of the study, may serve as a useful and accessible tool for identifying HMB in adolescent populations.

## Supporting information

**S1 File. Questionnaire.**
(DOCX)

**S1 Fig. Directed acyclic graph for identifying confounders between HMB and ID.**
(DOCX)

**S1 Table. General health in the study population, based on heavy menstrual bleeding (HMB) status.**
(DOCX)

**S2 Table. The use of iron supplements by menstrual status and dietary status, respectively.**
(DOCX)

**S3 Table. Levels of ferritin and hemoglobin, and rates of anemia and iron deficiency with cutoffs at ferritin<15 µg and <30 µg, respectively.** Data are presented overall and by dietary status and heavy menstrual bleeding (HMB).
(DOCX)

**S4 Table. Logistic regression analysis, crude and adjusted for BMI and years of menstruation, showing the odds ratio (OR) with 95% confidence interval (CI) for serum ferritin <30 µg/L.**
(DOCX)

**S5 Table. Rates of anemia (hemoglobin < 120g/L) among participants depending on heavy menstrual bleeding (HMB) status, BMI group and dietary preference. χ2 test or Fisher's exact test. Row percentages.**
(DOCX)

**S6 Table. Sensitivity analysis, excluding cases with imputed data.** N = 328 complete case analysis. Logistic regression analysis showing crude (odds ratio) OR and 95% confidence interval (CI) for ID (iron deficiency) and adjusted OR for years of menstruation and BMI.
(DOCX)

## Acknowledgments

The authors would like to thank Dr Stefan Lindgren for generously sharing his knowledge and expertise during the preparatory work for the study. We also wish to thank Professor Joaquin Calaf (University of Barcelona, Spain) for kindly allowing us to translate and use the SAMANTA questionnaire.

## Author contributions

**Conceptualization:** Anna Stubbendorff, Beata Borgström Bolmsjö, Moa Wolff.

**Data curation:** Lisa Söderman, Anna Stubbendorff.

**Formal analysis:** Lisa Söderman.

**Investigation:** Anna Stubbendorff, Beata Borgström Bolmsjö, Peter Nymberg, Moa Wolff.

**Methodology:** Lisa Söderman, Linnea V Ladfors.

**Project administration:** Moa Wolff.

**Resources:** Moa Wolff.

**Supervision:** Moa Wolff.

**Validation:** Lisa Söderman, Linnea V Ladfors, Moa Wolff.

**Visualization:** Lisa Söderman, Linnea V Ladfors.

**Writing – original draft:** Lisa Söderman.

**Writing – review & editing:** Lisa Söderman, Anna Stubbendorff, Linnea V Ladfors, Beata Borgström Bolmsjö, Peter Nymberg, Moa Wolff.

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
