## [Decision Letter · Decision Letter 0]

25 Aug 2025

Thank you for submitting your manuscript to PLOS ONE. After careful consideration, we feel that it has merit but does not fully meet PLOS ONE’s publication criteria as it currently stands. Therefore, we invite you to submit a revised version of the manuscript that addresses the points raised during the review process.

Please submit your revised manuscript by Oct 09 2025 11:59PM. If you will need more time than this to complete your revisions, please reply to this message or contact the journal office at plosone@plos.org . A rebuttal letter that responds to each point raised by the academic editor and reviewer(s). You should upload this letter as a separate file labeled 'Response to Reviewers'.A marked-up copy of your manuscript that highlights changes made to the original version. You should upload this as a separate file labeled 'Revised Manuscript with Track Changes'.An unmarked version of your revised paper without tracked changes. You should upload this as a separate file labeled 'Manuscript'.

We look forward to receiving your revised manuscript.

Kind regards,

Krzysztof Durkalec-Michalski, Ph.D

Academic Editor

PLOS ONE

Journal Requirements:

“Lisa Söderman and Moa Wolff received an honorarium for an educational webinar by Pharmacosmos.”

We note that one or more of the authors are employed by a commercial company: Pharmacosmos.

4. We noted in your submission details that a portion of your manuscript may have been presented or published elsewhere. [The in-depth nutritional and dietary aspects of this study have been presented in “Iron insight: exploring dietary patterns and iron deficiency among teenage girls in Sweden” by Stubbendorf et al. DOI: https://doi.org/10.1007/s00394-025-03630-z] Please clarify whether this [conference proceeding or publication] was peer-reviewed and formally published. If this work was previously peer-reviewed and published, in the cover letter please provide the reason that this work does not constitute dual publication and should be included in the current manuscript.

5. We note that you have indicated that there are restrictions to data sharing for this study. For studies involving human research participant data or other sensitive data, we encourage authors to share de-identified or anonymized data. However, when data cannot be publicly shared for ethical reasons, we allow authors to make their data sets available upon request. For information on unacceptable data access restrictions, please see http://journals.plos.org/plosone/s/data-availability#loc-unacceptable-data-access-restrictions.

6. In the online submission form you indicate that your data is not available for proprietary reasons and have provided a contact point for accessing this data. Please note that your current contact point is a co-author on this manuscript. According to our Data Policy, the contact point must not be an author on the manuscript and must be an institutional contact, ideally not an individual. Please revise your data statement to a non-author institutional point of contact, such as a data access or ethics committee, and send this to us via return email. Please also include contact information for the third party organization, and please include the full citation of where the data can be found.

Reviewers' comments:

Reviewer's Responses to Questions

**Comments to the Author**

1. Is the manuscript technically sound, and do the data support the conclusions?

Reviewer #1: Yes

Reviewer #2: Yes

Reviewer #3: Yes

2. Has the statistical analysis been performed appropriately and rigorously?

Reviewer #1: Yes

Reviewer #2: Yes

Reviewer #3: Yes

3. Have the authors made all data underlying the findings in their manuscript fully available?

Reviewer #1: Yes

Reviewer #2: No

Reviewer #3: Yes

4. Is the manuscript presented in an intelligible fashion and written in standard English?

Reviewer #1: Yes

Reviewer #2: Yes

Reviewer #3: Yes

Reviewer #1: The authors are commended for undertaking and expertly exploring the prevalence of iron deficiency in adolescents and how it relates to dietary habits and iron deficiency. Using a ferritin cutoff of < 15 to define iron deficiency (WHO recommendation)

As much as it can be inferred that a diet low in meat in addition to HMB can lead to greater iron deficiency, it is helpful to have this data and also apply the SAMANTA questionnaire.

Although there are recommendations to consume less meat for health/ethical reasons, this study raises the questions about how such recommendations can impact a cohort of adolescents at risk for iron deficiency.

Reviewer #2: Whilst the full dataset is not provided, a statement is made that it is accessible with caveats.

Title: Is ‘Iron insight’ needed at the beginning of the title?

General points:

This paper addresses an important area of health and a widely studied topic.

Good size cohort (n=394), from two schools.

Simple cross sectional study design describing prevalence and then Odds ratios calculated for iron deficiency by group and combined.

Findings not particularly novel but confirmatory of what has been reported elsewhere.

Very well written article with an excellent overview of the literature.

The article seems very similar to this article recently published where ferritin levels were reported according to diet types: Stubbendorff, A., Borgström Bolmsjö, B., Bejersten, T. et al. Iron insight: exploring dietary patterns and iron deficiency among teenage girls in Sweden. Eur J Nutr 64, 107 (2025). https://doi.org/10.1007/s00394-025-03630-z

The authors need to cite this article and make it clear how the present article is different.

I see that this is explained line 112-114, however, some further explanation would be welcomed, perhaps in the discussion.

Can the authors comment on how representative these school children are of the general population? i.e. the socio-demographic status?

I would recommend supplying the full questionnaires used, including the translations, as supplementary files.

Were samples transported to a laboratory or analysed onsite?

Please check the following statement, line 114:

“Those with anemia (Hb <107 g/L as per local reference values) or ferritin >150 μg/L were referred to their primary care center for follow-up.”

Should this be ferritin < 15 ug/L? If it is a threshold for iron overload, please provide an explanation.

Figure 2 typo: ’stainig’

Line 160-163 and Fig S1

Please explain how the DAG was constructed - what sources of information were used?

On the DAG figure itself, or caption, please provide more information, e.g. an explanation for the colours of the arrows.

Discussion

LIne 288-291. I am somewhat concerned about the suggestion of increasing use of iron replacement therapy, tranexamic acid or hormonal contraceptives in this young population. These come with their own risks and this must be acknowledged here. A decision would need to be made - which one carries a higher health risk? HC use in particular could be counter-productive as it removes the natural health signal of menstruation which has been termed a ‘vital sign’, and carries other health risks. This needs a fuller explanation or should be removed. At the very least non-pharmacological interventions should be considered/attempted first in these young females.

Line 350: There is substantial evidence that menstruating females require more iron than males due to menstrual blood losses. Therefore I would suggest that a lower cutoff for females doesn’t lack scientific evidence, in fact it runs counter to the scientific evidence.

Reviewer #3: Dear Editor, Dear Authors,

Thank you for the opportunity to review the manuscript titled: “Iron insight - Exploring the effect of menstrual loss and dietary iron intake on iron deficiency in teenagers: a cross-sectional study”. The results of this study identify the important problem of iron deficiency of adolescent girls. The manuscript is well written, however I have a few questions and suggestions:

1. The title of the study states, that it “explores dietary iron intake”, however from what I understand from the methods section, this particular manuscript is based on participant self-report of meat restriction diet. There is no information on dietary assessment methods used, therefore we do not know what is the dietary iron intake among the participants. As described in the discussion, the intake of iron itself could be within dietary recommendations, however combining dietary factors, such as consuming vitamin C, phytates, or casein could influence the absorption.

2. There is no information on the toxicity of supplemental iron, which I would recommend to add in the discussion. Are there any food products mandatory fortified with iron in Sweden?

3. For future consideration, it would be interesting to collect information on physical activity of the participants; regular physical activity seem to reduce risk of menstruation irregularities, including the heavy menstrual bleeding.

**Do you want your identity to be public for this peer review?** For information about this choice, including consent withdrawal, please see our Privacy Policy

Reviewer #1: No

Reviewer #2: No

Reviewer #3: No

---

## [Author Response · Author response to Decision Letter 1]

11 Oct 2025

Journal Requirements:

Thank you for noticing! We have made the appropriate adjustments to the style.

Thank you for your comment. In Sweden, parental consent is not required for participants aged 15 years or older. This information has now been added to the Methods section (line 93).

“Lisa Söderman and Moa Wolff received an honorarium for an educational webinar by Pharmacosmos.”

We note that one or more of the authors are employed by a commercial company: PharmacosmosK.

Thank you for the guidance. The Funding Statement has been amended to clarify that there is no employment relationship with the company and that the honorarium was only for the educational webinar, unrelated to the present study. The specific roles of the authors are described in the Author Contributions section.

Thank you for carefully reviewing the Competing Interests statement. At one single occasion, two of the co-authors gave digital educational lectures aimed at healthcare professionals – one on iron deficiency in adolescents (MW) and the other on menstruation an iron deficiency (LS). For these webinars, they each received a one-time honorarium from Pharmacosmos. However, there is no employment relationship with the company, and these activities were unrelated to the present study. This has now been further clarified in the Competing Interests section.

We have added the statement: This does not alter our adherence to PLOS ONE policies on sharing data and materials.

Both the updated Funding Statement and the updated Competing Interests Statement have been included in the cover letter, as requested. Thank you for your guidance.

4. We noted in your submission details that a portion of your manuscript may have been presented or published elsewhere. [The in-depth nutritional and dietary aspects of this study have been presented in “Iron insight: exploring dietary patterns and iron deficiency among teenage girls in Sweden” by Stubbendorf et al. DOI: https://doi.org/10.1007/s00394-025-03630-z] Please clarify whether this [conference proceeding or publication] was peer-reviewed and formally published. If this work was previously peer-reviewed and published, in the cover letter please provide the reason that this work does not constitute dual publication and should be included in the current manuscript.

Thank you for raising this important point. We would like to clarify that there is no dual publication.

The previously published article by Stubbendorff et al. focused exclusively on dietary patterns and iron deficiency. In that paper, menstrual data were only included as a confounding variable to adjust the analyses of the relationship between diet and iron deficiency. In contrast, the current manuscript specifically investigates menstrual factors in relation to iron deficiency and provides an in-depth analysis and discussion of these factors, which were not addressed in the previous publication.

Therefore, the two manuscripts address distinct research questions and make separate scientific contributions. We have also clarified this distinction in the cover letter to ensure transparency for the editors and reviewers.

5. We note that you have indicated that there are restrictions to data sharing for this study. For studies involving human research participant data or other sensitive data, we encourage authors to share de-identified or anonymized data. However, when data cannot be publicly shared for ethical reasons, we allow authors to make their data sets available upon request. For information on unacceptable data access restrictions, please see http://journals.plos.org/plosone/s/data-availability#loc-unacceptable-data-access-restrictions.

Based on the regulations outlined in the approved ethics application, it is not possible to make the data publicly available. Participants were informed that results would only be published at the group level and could not be traced back to individuals. Making the data openly accessible to other researchers could allow identification of individual-level information and would therefore breach compliance with the protocol approved by the Swedish Ethical Review Authority. However, data can be made available upon specific request and arrangements.

6. In the online submission form you indicate that your data is not available for proprietary reasons and have provided a contact point for accessing this data. Please note that your current contact point is a co-author on this manuscript. According to our Data Policy, the contact point must not be an author on the manuscript and must be an institutional contact, ideally not an individual. Please revise your data statement to a non-author institutional point of contact, such as a data access or ethics committee, and send this to us via return email. Please also include contact information for the third party organization, and please include the full citation of where the data can be found.

The data underlying our study have now been uploaded to the Swedish National Data Service (SND) repository and are currently under review. The dataset will be available upon reasonable request via request@snd.se after publication.

-

The reference list has been reviewed.

Reviewers' comments:

Reviewer's Responses to Questions

Comments to the Author

1. Is the manuscript technically sound, and do the data support the conclusions?

Reviewer #1: Yes

Reviewer #2: Yes

Reviewer #3: Yes

2. Has the statistical analysis been performed appropriately and rigorously?

Reviewer #1: Yes

Reviewer #2: Yes

Reviewer #3: Yes

3. Have the authors made all data underlying the findings in their manuscript fully available?

Reviewer #1: Yes

Reviewer #2: No

Reviewer #3: Yes

4. Is the manuscript presented in an intelligible fashion and written in standard English?

Reviewer #1: Yes

Reviewer #2: Yes

Reviewer #3: Yes

5. Review Comments to the Author

Reviewer #1: The authors are commended for undertaking and expertly exploring the prevalence of iron deficiency in adolescents and how it relates to dietary habits and iron deficiency. Using a ferritin cutoff of < 15 to define iron deficiency (WHO recommendation)

As much as it can be inferred that a diet low in meat in addition to HMB can lead to greater iron deficiency, it is helpful to have this data and also apply the SAMANTA questionnaire.

Although there are recommendations to consume less meat for health/ethical reasons, this study raises the questions about how such recommendations can impact a cohort of adolescents at risk for iron deficiency.

Reviewer #2: Whilst the full dataset is not provided, a statement is made that it is accessible with caveats.

Title: Is ‘Iron insight’ needed at the beginning of the title?

Thank you for your comment. The phrase ‘Iron insight’ has been removed from the title to improve clarity.

General points:

This paper addresses an important area of health and a widely studied topic.

Good size cohort (n=394), from two schools.

Simple cross sectional study design describing prevalence and then Odds ratios calculated for iron deficiency by group and combined.

Findings not particularly novel but confirmatory of what has been reported elsewhere.

Very well written article with an excellent overview of the literature.

Thank you!

The article seems very similar to this article recently published where ferritin levels were reported according to diet types: Stubbendorff, A., Borgström Bolmsjö, B., Bejersten, T. et al. Iron insight: exploring dietary patterns and iron deficiency among teenage girls in Sweden. Eur J Nutr 64, 107 (2025). https://doi.org/10.1007/s00394-025-03630-z

The authors need to cite this article and make it clear how the present article is different.

I see that this is explained line 112-114, however, some further explanation would be welcomed, perhaps in the discussion.

The previously published article by Stubbendorff et al. focused exclusively on dietary patterns and iron deficiency. In that paper, menstrual data were only included as a confounding variable to adjust the analyses of the relationship between diet and iron deficiency. In contrast, the current manuscript specifically investigates menstrual factors in relation to iron deficiency and provides an in-depth analysis and discussion of these factors, which were not addressed in the previous publication. This is now clarified in the Discussion section, line 299.

Can the authors comment on how representative these school children are of the general population? i.e. the socio-demographic status?

Thank you for this valuable comment. To better illustrate the representativeness of the cohort, we have added the socio-demographic variable urban/small town/rural to Table 1. This provides an overview of the participants’ backgrounds and clarifies the distribution between those from urban areas, small towns, and rural areas.

I would recommend supplying the full questionnaires used, including the translations, as supplementary files.

Thank you for this helpful suggestion. The full questionnaires used, including the translations, have now been added as supplementary information.

Were samples transported to a laboratory or analysed onsite?

Thank you for your question. The samples were transported to a laboratory, and this information has now been clarified in the Methods section (line 135).

Please check the

---

## [Decision Letter · Decision Letter 1]

30 Oct 2025

Exploring the effect of menstrual loss and dietary iron intake on iron deficiency in teenagers: a cross-sectional study

PONE-D-25-27936R1

Dear Dr. Söderman

We’re pleased to inform you that your manuscript has been judged scientifically suitable for publication and will be formally accepted for publication once it meets all outstanding technical requirements.

Kind regards,

Krzysztof Durkalec-Michalski, Ph.D

Academic Editor

PLOS ONE

Additional Editor Comments (optional):

Reviewers' comments:

Reviewer's Responses to Questions

**Comments to the Author**

Reviewer #1: All comments have been addressed

Reviewer #2: All comments have been addressed

Reviewer #3: All comments have been addressed

2. Is the manuscript technically sound, and do the data support the conclusions?

Reviewer #1: Yes

Reviewer #2: Yes

Reviewer #3: Yes

3. Has the statistical analysis been performed appropriately and rigorously?

Reviewer #1: Yes

Reviewer #2: Yes

Reviewer #3: Yes

4. Have the authors made all data underlying the findings in their manuscript fully available?

Reviewer #1: Yes

Reviewer #2: (No Response)

Reviewer #3: Yes

5. Is the manuscript presented in an intelligible fashion and written in standard English?

Reviewer #1: Yes

Reviewer #2: Yes

Reviewer #3: Yes

Reviewer #1: I appreciate that the authors have taken the time to address the concerns that I had initially and should be commended. Thank you very much.

Reviewer #2: I am satisfied that the reviewer comments have been satisfactorily addressed and I have no further comments.

Reviewer #3: Thank you for thoroughly adressing all the comments! Here are some studies which analyse the association between HMB and physical activity: doi: 10.3390/healthcare12192005; doi: 10.1093/humrep/deab055. Sweden has interesting history of iron fortification: doi: 10.1093/humrep/deab055.

**Do you want your identity to be public for this peer review?** For information about this choice, including consent withdrawal, please see our Privacy Policy

Reviewer #1: No

Reviewer #2: No

Reviewer #3: No

---

## [Editor Report · Acceptance letter]

PONE-D-25-27936R1

PLOS ONE

Dear Dr. Söderman,

I'm pleased to inform you that your manuscript has been deemed suitable for publication in PLOS ONE. Congratulations! Your manuscript is now being handed over to our production team.

Kind regards,

on behalf of

Dr. Krzysztof Durkalec-Michalski

Academic Editor

PLOS ONE